# Exercise Inhibits Doxorubicin-Induced Damage to Cardiac Vessels and Activation of Hippo/YAP-Mediated Apoptosis

**DOI:** 10.3390/cancers13112740

**Published:** 2021-06-01

**Authors:** Rong-Hua Tao, Masato Kobayashi, Yuanzheng Yang, Eugenie S. Kleinerman

**Affiliations:** 1Department of Pediatrics-Research, Division of Pediatrics, The University of Texas M.D. Anderson Cancer Center, Houston, TX 77030, USA; YYang9@mdanderson.org; 2School of Health Sciences, Institutes of Medical, Pharmaceutical and Health Sciences, Kanazawa University, Kanazawa 920-0942, Japan; kobayasi@mhs.mp.kanazawa-u.ac.jp

**Keywords:** doxorubicin, cardiotoxicity, exercise, BM stem cells, endothelial cells, pericytes, Hippo-YAP signaling, apoptosis

## Abstract

**Simple Summary:**

We developed a juvenile mouse cardiotoxicity model and demonstrated that doxorubicin treatment led to cardiac vascular damage involving a decrease in vascular endothelial cells and pericytes. By contrast, cardiac vessels in mice treated with exercise (Ex) during Dox treatment had no change in vascular endothelial cells or pericyte coverage. Here we show that exercise during Dox induced the migration of bone marrow (BM) stem cells into the heart with their differentiation into vascular endothelial cells and pericytes. These data define a new mechanism of Dox-induced cardiotoxicity and suggest that Ex inhibits vascular damage by promoting bone marrow (BM) stem cell-mediated cardiac vessel repair. We also show that Ex inhibits Hippo-YAP signaling-mediated apoptosis in cardiomyocytes. Ex may be an effective intervention strategy. Targeting Hippo-YAP signaling may also offer another novel therapeutic option to protect the heart.

**Abstract:**

Dose-related cardiomyopathy is a major side effect following doxorubicin (Dox). To investigate whether exercise (Ex)-induced vasculogenesis plays a role in reducing Dox-induced cardiotoxicity, GFP^+^ bone marrow (BM) cells from GFP transgenic mice were transplanted into wild-type mice. Transplanted mice were treated with Dox, Ex, Dox+Ex, or control. We found Dox therapy resulted in decreased systolic and diastolic blood flow, decreased ejection fraction and fractional shortening, and decreased vascular endothelial cells and pericytes. These abnormalities were not seen in Dox+Ex hearts. Heart tissues from control-, Ex-, or Dox-treated mice showed a small number of GFP^+^ cells. By contrast, the Dox+Ex-treated hearts had a significant increase in GFP^+^ cells. Further analyses demonstrated these GFP^+^ BM cells had differentiated into vascular endothelial cells (GFP^+^CD31^+^) and pericytes (GFP^+^NG2^+^). Decreased cardiomyocytes were also seen in Dox-treated but not Dox+Ex-treated hearts. Ex induced an increase in GFP^+^c-Kit^+^ cells. However, these c-Kit^+^ BM stem cells had not differentiated into cardiomyocytes. Dox therapy induced phosphorylation of MST1/2, LATS1, and YAP; a decrease in total YAP; and cleavage of caspase-3 and PARP in the heart tissues. Dox+Ex prevented these effects. Our data demonstrated Dox-induced cardiotoxicity is mediated by vascular damage resulting in decreased cardiac blood flow and through activation of Hippo-YAP signaling resulting in cardiomyocyte apoptosis. Furthermore, Ex inhibited these effects by promoting migration of BM stem cells into the heart to repair the cardiac vessels damaged by Dox and through inhibiting Dox-induced Hippo-YAP signaling-mediated apoptosis. These data support the concept of using exercise as an intervention to decrease Dox-induced cardiotoxicity.

## 1. Introduction

Doxorubicin (Dox) is one of the most effective chemotherapy agents for treating children, adolescents, and young adults with bone and soft-tissue sarcomas [1]. However, Dox-induced cardiovascular disease and heart failure, which compromise both quality of life (QOL) and long-term survival, are the most common late effects seen in childhood cancer survivors [2,3]. The frequency of cardiovascular dysfunction after Dox treatment has been reported to be between 57% and 65%, more than 5 times higher than in community controls. Dox-induced cardiotoxicity is the second-leading cause of death after cancer relapse. When congestive heart failure develops, the mortality rate is approximately 50% [4,5,6]. Mechanisms of Dox-induced cardiotoxicity include the generation of excess reactive oxygen species (ROS) that damage mitochondria, iron complex formation, and inhibition of topoisomerase-IIβ (Top2β), all of which result in cardiomyocyte apoptosis [1,7,8,9,10,11]. This myocardial apoptosis results in cardiac dysfunction and heart failure [12,13,14,15,16]. Although recent studies in children with leukemia showed decreased cardiotoxicity in patients treated with dexrazoxane, this agent only targets ROS and has not been definitively studied in sarcoma patients, who typically receive much higher doses of Dox (450 mg/m^2^ versus 250 mg/m^2^) [17]. Further investigation of the molecular pathways responsible for Dox-induced cardiomyocyte apoptosis is needed to identify effective interventions.

We recently described a juvenile mouse Dox-induced cardiotoxicity model and showed that Dox-treated mice had compromised cardiac function immediately after therapy, late diastolic failure, and cardiac vessel abnormalities consisting of decreased vascular endothelial cells and pericytes [18]. Decreased pericyte coverage results in closed vascular lumens, decreased functionality, increased vessel leakage, and ineffective oxygen delivery, suggesting that the vascular compromise induced by Dox may contribute to cardiomyocyte apoptosis. We also concluded that the Dox-induced effect on pericyte coverage in the cardiac vessels may result in decreased cardiac blood flow. Using this model, we demonstrated that exercise (Ex) during or after Dox treatment significantly reduced both early and late cardiotoxicity, including the cardiac vessel changes, without affecting tumor response [18]. The cardiac vessels in the exercised mice showed no Dox-induced changes in endothelial or pericyte coverage and were similar to those of the normal controls. The mechanism of vascular preservation in the Dox+Ex-treated mice was the focus of this present study. 

We previously used a transplant model with GFP-labeled bone marrow (BM) stem cells to investigate the contribution of BM cells to new vessel formation. We demonstrated that GFP-positive BM cells contributed to new tumor vessel formation by migrating into the tumor and differentiating into vascular endothelial cells and pericytes [19,20,21,22,23,24]. Blocking BM cell differentiation into vascular cells severely inhibited vascular expansion. To ascertain whether exercise induces the migration of BM stem cells into the heart, where they contribute to cardiac vascular repair and the preservation of cardiac blood flow, we once again employed this BM transplant model.

In addition, we investigated whether the Hippo-YAP pathway plays a role in Dox-induced cardiomyocyte apoptosis. The Hippo signaling cascade plays an evolutionarily conserved role in organ size and growth by regulating cell differentiation, proliferation, and apoptosis [25,26,27,28]. The core of the Hippo signaling pathway comprises the serine/threonine kinases mammalian STE20-like protein kinase 1 and 2 (MST1 and MST2) and large tumor suppressor homolog 1 and 2 (LATS1 and LATS2). The main function of this pathway is to suppress the activity of its downstream effectors Yes-associated protein (YAP) and transcriptional coactivator with PDZ-binding motif (TAZ). YAP and TAZ promote cell migration, differentiation, proliferation, and survival and inhibit apoptosis. Recent findings suggest that the Hippo-YAP signaling pathway can be redeployed to promote heart regeneration and repair [25,26,27,28]. Activation of Hippo signaling by expression of wild-type Mst1 promoted cardiomyocyte apoptosis, impaired heart function, and caused mouse death [29]. Suppression of Hippo signaling by overexpression of a dominant-negative mutant of Mst1 in the heart did not significantly affect heart function but improved myocardial outcome after myocardial infarction (MI). In these hearts, expressing the dominant-negative mutant of Mst1 reduced cardiomyocyte apoptosis, decreased fibrosis, and preserved systolic contraction post-MI [29,30]. These experiments showed that activation of Hippo signaling promotes apoptosis in cardiomyocytes [26]. Since Hippo signaling promotes cardiomyocyte apoptosis and Dox induces myocardial apoptosis, we investigated whether Dox activated the Hippo signaling pathway in cardiomyocytes. We are the first to show that Dox-induced cardiotoxicity may be mediated through activation of the Hippo-YAP signaling pathway and that exercise prevents or inhibits activation of this pathway.

This study identifies two new mechanisms of Dox-induced cardiotoxicity, i.e., cardiac vascular damage and activation of Hippo-YAP signaling, and also uncovers the mechanism by which Ex suppresses Dox-induced cardiotoxicity, thereby providing additional new targets for intervention with the goal of decreasing cardiac late effects in survivors. It also confirms our previous studies showing that Ex during or after Dox treatment may be a novel intervention strategy for sarcoma and other patients treated with Dox.

## 2. Methods

All experimental materials, analytic methods, and supporting data are available in the article and its online supplementary files. An expanded Methods section can be found in the online-only Data Supplement. The information of key resources of the primary and secondary antibodies used in this study can be found in Tables in the online-only Data Supplement.

### 2.1. Mice

All mouse procedures were performed according to institutional guidelines and governmental regulations. All mouse experimental procedures were approved by the Institutional Animal Care and Use Committee (IACUC) at The University of Texas MD Anderson Cancer Center under protocol number IACUC00001615-RN00. Recipient mice of strain C57BL/6 were purchased at 4 weeks old from the Animal Facility, Experimental Radiation Oncology, MD Anderson Cancer Center, Houston, TX. Donor GFP transgenic mice of strain C57BL/6-Tg(UBC-GFP)30Scha/J were purchased from The Jackson Laboratory.

### 2.2. Cell Culture

Human cardiac myocytes (HCMs) were cultured with cardiac myocyte medium (CMM) in poly-L-lysine-coated dishes and supplemented with 5% fetal bovine serum (FBS), 1% cardiac myocyte growth supplement (CMGS), and 1% penicillin/streptomycin solution (P/S) (all were purchased from ScienCell Research Laboratories, Inc., Carlsbad, CA, USA). Cells were maintained in an incubator with 5% CO_2_ atmosphere at 37 °C.

### 2.3. Bone Marrow Transplant (BMT)

Whole BM cells were collected by flushing femurs from donor GFP transgenic mice with phosphate-buffered saline (PBS). Recipient C57BL/6 mice were treated intraperitoneally with 25 mg/kg of busulfan (Cat. #B2635; Sigma, St. Louis, MO, USA) 4 days before BMT and with 30 µg of anti-CD4 (Clone GK1.5, Cat. #BE0003-1, Bio-X-Cell, Lebanon, NH, USA) and anti-CD8 (Clone 2.43, Cat. #BE0061, Bio-X-Cell) antibodies 2 days before BMT. Treated mice were then injected intravenously with 5 × 10^6^ GFP**^+^** BM cells. After allowing 4 weeks for BM cell engraftment, the transplant efficiency was confirmed by evaluating GFP**^+^** BM cells collected from at least 3 representative transplanted mice using flow cytometry (BD FACSCalibur, BD Biosciences, San Jose, CA, USA; Appendix A).

### 2.4. Exercise (Ex) and Doxorubicin (Dox) Treatment

Treadmill Exercise Protocol: Mice were exercised 5 or 6 days a week with 1 or 2 days off for a total of 2 weeks (total 10 or 12 days of exercise) as indicated. Mice were placed on a treadmill with an operating protocol consisting of 45 min of constant walking at a rate of 12 m/min at 0% slope with 2 min of ramp-up from 0 m/min and 2 min of cooldown to 0 m/min at the beginning and end of exercise, respectively, as reported previously [7]. When exercise was completed, the mice were immediately returned to their cages and allowed to act freely.

Doxorubicin Treatment: Mice were injected intravenously with 2.5 mg/kg of Dox twice a week for 2 weeks. Dox injection was administered on days 3, 5, 10, and 12 after the start of the exercise protocol, as illustrated in Figure 1A and Figure 3A.

### 2.5. Echocardiographic Imaging

Before and after bone marrow transplantation and Dox/Ex treatment, mouse cardiac function and morphology were monitored under anesthetization with 1% isoflurane when the heart rate reached around 425 bpm using a Vevo 2100 High Resolution Imaging System equipped with a 30 MHz probe (MS400) (FUJIFILM VisualSonics, Inc., Toronto, ON, Canada), as described previously [7,31]. Mouse echocardiographic functional and morphological parameters were measured and analyzed using the parasternal long axis or short axis M-mode and Doppler imaging with Vevo LAB 3.1.1 software provided by the system’s manufacturer (FUJIFILM VisualSonics, Inc.) and GraphPad Prism 6 software (GraphPad Software, Inc., San Diego, CA, USA).

### 2.6. Immunohistochemical Analysis

Frozen slides and H&E staining were prepared and carried out by the MD Anderson Research Histology Core Laboratory. Fixation and permeabilization of frozen tissues were performed by submersion in cold acetone for 10 min as described previously [32]. Images were captured with a Leica DM5500 B upright microscope imaging system (Leica Microsystems, Buffalo Grove, IL, USA) and analyzed using Adobe Photoshop software (Adobe Inc., San Jose, CA, USA).

### 2.7. Confocal Laser Scanning Microscope

The high magnification imaging and Z-series stack were performed in the Flow Cytometry & Cellular Imaging Facility at MD Anderson Cancer Center. Stained slides were visualized under an FV1000 confocal laser scanning microscope with ISS upgrade (Olympus Corp., Shinjuku City, Tokyo, Japan), and images were captured using FV10-ASW 4.0 software (Olympus Corp.). Images were acquired sequentially to minimize cross-contamination from multiple emission spectra. Identical settings for laser intensity and other image capture parameters were applied for comparison of staining from different mouse groups [32,33]. Examination of signal intensity and reconstitution of Z-series stacks into 3D volume viewer, 3D surface plot, and videos were performed using ImageJ software (National Institutes of Health, Bethesda, MD, USA).

### 2.8. MTT Assay

HCMs were cultured in 96-well microplates and treated with various concentrations of Dox for the time periods indicated in the figures. After labeling with 3-(4,5-dimethyl-2-thiazolyl)-2,5-diphenyl-2H-tetrazolium bromide (MTT/thiazolyl blue tetrazolium bromide, Sigma), cell growth was monitored using a SpectraMax Plus 384 microplate reader (Molecular Devices, LLC, San Jose, CA, USA). Data were analyzed using Excel [34].

### 2.9. Western Blot Analysis

Treated HCMs and mouse heart tissues were collected, and cell extracts were prepared for Western blot analysis with the indicated antibodies under conditions as described in previous reports [35,36]. The original Western Blots are shown in Appendix A.

### 2.10. Statistical Analysis

Statistical analysis of echocardiographic data was performed using the GraphPad *t* test. For other data analysis, Student’s *t* test was applied. Data are presented as mean ± SEM, and *p* < 0.05 was considered statistically significant.

## 3. Results

### 3.1. Exercise Prevents Dox-Induced Decreases in Cardiac Function and Blood Flow

We previously demonstrated that Ex during Dox therapy prevented acute cardiac vascular damage as defined by decreased vascular endothelial cells and pericytes [18]. Since impaired pericyte vascular coverage will result in poorly perfused vessels and decreased blood flow, we quantified diastolic and systolic blood flow in Dox- and Dox+Ex-treated mice. Mice were treated with Dox, Dox+Ex, or control (no Dox and no Ex) for 2 weeks. Echocardiography (Echo) was performed prior to and 24 h after therapy (Figure 1A). In addition to a significant decrease in left ventricular ejection fraction (LVEF) and left ventricular fractional shortening (LVFS), mice treated with Dox showed decreased cardiac mean velocity at left ventricular mitral valve (LVMV) and at left ventricular outflow tract (LVOT) in the ascending aorta, indicating impaired diastolic and systolic blood flow (Figure 1B–H). No changes in LVEF, LVFS, or measures of blood flow were seen in the mice treated with Dox+Ex compared to controls (Figure 1B–H). Ex alone had no significant effect on cardiac function and blood flow (Figure 1I–K). We hypothesized that the normal cardiac blood flow in the Dox+Ex mice was secondary to the preservation or restoration of the endothelial and pericyte layers in the Dox-damaged cardiac vessels. 

We previously showed that BM stem cells can migrate and participate in vascular expansion and repair by differentiating into endothelial cells and pericytes that contribute to the formation of the new vessels [19,20,21,22,23,24]. To determine whether this process is involved in the rescue of damaged cardiac vessels following Dox treatment, we performed a BMT using GFP^+^ BM cells isolated from the femurs of GFP transgenic mice (Figure 2A). This allowed us to determine whether some/all of the endothelial cells and pericytes in the cardiac vessels were derived from BM cells, as these cells were GFP-positive. First, we investigated the effect of BMT on cardiac function (Figure 2B). Echo data showed that there were no significant differences in LVEF, LVFS, mitral valve (MV) E/A ratio, MV ejection time (ET), LV anterior/posterior wall thickness in diastole (LVAW;d/LVPW;d) (Figure 2C–H), LV internal diameter in systole (LVID;s), LV volume in systole (LV Vol.;s), LV mass, isovolumic contraction time (IVCT), isovolumic relaxation time (IVRT), and myocardial performance index (MPI) (Appendix A) between control and BMT mice. These findings indicated that there was no significant adverse cardiac effect secondary to the transplant. Next, transplanted mice were randomly grouped and treated with Dox, Ex, Dox+Ex, or control. Cardiac function was assessed by Echo, and heart tissues were collected for analysis (Figure 2A and Figure 3A). Dox treatment significantly decreased LVEF and LVFS (Figure 3B,C) and increased LVAW;d, LVPW;d, heart weight to tibia length/body weight (HW/TL, HW/BW), LVID;s, LV Vol.;s, LV mass, IVCT, IVRT, and MPI (Appendix A). By contrast, there were no significant differences between the control and Dox+Ex groups. These data confirmed the preservation of cardiac function in the Dox+Ex transplanted mice. 

### 3.2. Exercise Inhibits Dox-Induced Cardiomyocyte Apoptosis 

Dox has been shown to induce apoptosis of cardiomyocytes [37,38,39,40,41]. We first examined the cardiac tissue, specifically focusing on the cardiomyocytes using immunofluorescent staining with cardiac troponin I (cTn I). Compared to control hearts, there was a significant decrease in cTn I^+^ cells in the Dox-treated hearts (Figure 3D). Dox also induced cleavage of caspase-3 and PARP in the heart tissues (Figure 3E,F), indicating cardiomyocyte apoptosis. By contrast, cTn I, cleaved caspase-3, and cleaved PARP staining in the hearts from the Dox+Ex mice were similar to the control group. These data suggested that Ex protected the cardiomyocytes from Dox-induced apoptosis, resulting in the preservation of the cardiomyocyte numbers.

### 3.3. Exercise Promotes Migration of Bone Marrow (Stem) Cells into the Heart after Dox Treatment

Mouse hearts are composed of ~56% cardiomyocytes, 27% fibroblasts, 7% endothelial cells, and 10% pericytes and immune cells [42]. To determine whether BM stem cells participated in Ex-induced repair of the cardiac vasculature and cardiomyocytes damaged by Dox, we performed immunofluorescent staining analysis of the cardiac tissue. Migrated BM cells were identified using GFP staining. There were a small number of GFP^+^ cells in heart tissues from control-, Ex-, or Dox-treated mice (Figure 4A,B, Appendix A). In contrast, a significantly higher number of GFP^+^ cells were seen in the heart tissues from mice treated with Dox+Ex (Figure 4B, Appendix A). The GFP positivity of these cells in the heart indicated that they migrated from the BM. However, all transplanted BM cells and their derived cells are GFP-positive, and these cells are quite different in size and shape. So, GFP staining looked quite ununiform. Moreover, staining for GFP cannot identify stem cells or inform whether differentiation to a specific cell type occurred following migration.

We next used c-Kit to identify the total number of stem cells and GFP/c-Kit double staining to identify stem cells that originated from the BM. As shown in Figure 4C–E and Appendix A, there was a significant increase in the number of GFP and c-Kit double-positive cells in the hearts from Dox+Ex mice. Quantification of the dual immunofluorescent cells indicated that many c-Kit^+^ cells in the hearts of Dox+Ex mice were also GFP-positive (Figure 4D,E and Appendix A). These data indicate that Ex increased c-Kit^+^ stem cells in the heart after Dox treatment, and these c-Kit^+^ stem cells originated from the BM. To further confirm that the double-positive cells expressed both GFP and c-Kit, these cells were examined for GFP and c-Kit signals under 400× magnification (Appendix A) and also subjected to Z-stack imaging, which was then reconstituted into a 3D volume viewer and 3D surface plot (Appendix A). These data confirmed that the double-positive cells expressed both GFP and c-Kit (with GFP locating in the cytoplasm and c-Kit on the cell membrane) and indicated that these double-positive stem cells originated from BM.

Since Ex increased BM stem cells (GFP^+^/c-Kit^+^) in the heart after Dox treatment and preserved the number of cardiomyocytes, we investigated whether these BM stem cells differentiated into cardiomyocytes using dual immunofluorescent staining of both GFP and cTn I. As shown in Appendix A, few GFP and cTn I double-positive cells were found in the heart tissues from Dox+Ex mice, suggesting that the GFP^+^/c-Kit^+^ BM-derived stem cells were not differentiating into cardiomyocytes.

### 3.4. Exercise Promotes Differentiation of BM Stem Cells into Endothelial Cells and Pericytes after Dox Treatment

We next investigated whether preservation of blood flow, as well as endothelial cells and pericytes, in the cardiac vessels was due to the differentiation of migrated BM stem cells into vascular cells in the Dox+Ex hearts. As shown in Figure 5A,B, endothelial cells were significantly decreased in the heart tissues from Dox-treated mice compared to controls. By contrast, there was no significant difference in endothelial cells between control and Dox+Ex mice. To determine whether the migrated BM stem cells in the Dox+Ex hearts had differentiated into endothelial cells, dual immunofluorescent staining for both GFP and CD31 was performed. When compared to the control and Ex groups, the number of CD31^+^ cells was significantly lower in the Dox-treated but not the Dox+Ex-treated hearts (Figure 5C). As demonstrated above (Figure 4A,B), the number of GFP^+^ cells was significantly increased in the Dox+Ex hearts (Figure 5C). The colocalization of GFP and CD31 indicated that a portion of these CD31^+^ cells were derived from BM cells (Figure 5C and Appendix A). In Dox+Ex-treated mouse hearts, the total number and percentage of CD31^+^/GFP^+^ cells were significantly higher than in control-, Ex-, and Dox-treated mice (Figure 5D,E and Appendix A). These data suggested that Ex promoted the migration of BM stem cells into the heart and their differentiation into endothelial cells. To further confirm that these colocalized cells were expressing both GFP and CD31, high magnification images were taken using a confocal laser scanning microscope. Appendix A shows several GFP/CD31 double-positive cells in Dox+Ex-treated mouse hearts. Z-stack imaging and its reconstituted 3D volume viewer and 3D surface plot confirmed the presence of both GFP and CD31 in the same cells, with GFP locating in the cytoplasm and CD31 locating on the cell membrane (Appendix A, video S3, and video S4). These data indicate that the double-positive cells were endothelial cells that had differentiated from BM stem cells.

Next, we examined the effect of Dox therapy on cardiac pericytes using NG2 staining. Compared to the hearts from control and Ex mice, cardiac pericytes were significantly decreased in heart tissues from Dox-treated but not Dox+Ex-treated mice (Figure 6A,B). To determine whether BM stem cells participated in the recovery of pericytes in the hearts of Dox+Ex mice, we performed double immunofluorescence staining for both GFP and NG2. As shown in Figure 6C–E and Appendix A, in Dox+Ex hearts, there was a significant increase in GFP^+^ cells that also expressed NG2 when compared to the control-, Ex-, and Dox-treated hearts. This colocalization confirms that GFP^+^ pericytes were derived from the BM cells that migrated into the heart. The total number and percentage of pericytes derived from BM cells were significantly increased in Dox+Ex hearts (Figure 6D,E and Appendix A). These data suggested that Ex promoted the migration of BM stem cells into the heart with their subsequent differentiation into pericytes. High-magnification images by a confocal laser scanning microscope were used to confirm coexpression of GFP and NG2. Appendix A shows the GFP/NG2 double-positive cell in Dox+Ex-treated mouse hearts under 100× and 500× magnification. Z-stack imaging and its reconstituted 3D volume viewer and 3D surface plot confirmed the presence of both GFP and NG2 in the same cells, with GFP locating in the cytoplasm and NG2 locating on the cell membrane (Appendix A). These results indicate that a portion of the pericytes in the Dox+Ex hearts were derived from the migrated BM cells. 

We next evaluated whether Dox therapy resulted in compromised cardiac blood flow in the BM-transplanted mice and the effect of combining Dox+Ex. As demonstrated in nontransplanted mice (Figure 1B–H), Dox treatment resulted in a significant decrease in cardiac blood flow as quantified by the MV E/A Ratio and MV ET (Appendix A). By contrast, there were no significant differences in blood flow between the control and Dox+Ex transplanted mice.

### 3.5. Effect of Dox and Dox+Ex on Cardiac Fibroblast Proliferation

We next used vimentin staining to evaluate the effect of Dox versus Dox+Ex on cardiac fibroblasts. There was a significant increase in vimentin-positive cells following Dox therapy, indicating fibroblast proliferation. This was not seen in the control, Ex, or Dox+Ex hearts (Figure 7A). Co-localization of GFP and vimentin revealed very few double-positive cells in these hearts (Figure 7B). These data indicate that Dox therapy induced the proliferation of cardiac fibroblasts. However, these vimentin-positive cells were not derived from BM stem cells. Ex during therapy prevented the Dox-induced increase in cardiac fibroblasts.

### 3.6. Dox-Induced Cardiomyocyte Apoptosis is Mediated by Activation of Hippo-YAP Signaling

We demonstrated that Ex during Dox therapy inhibited cardiomyocyte apoptosis (Figure 3D–F). We also showed that BM stem cells played no role in preserving or replacing the damaged cardiomyocytes (Appendix A). While our data indicated that the decrease in cardiomyocyte apoptosis was due to the contribution of BM stem cells in repairing damaged cardiac vessels and thereby preserving blood flow, we also investigated possible molecular mechanisms responsible for the induction of the apoptosis process.

The Hippo-YAP pathway has been shown to control cardiomyocyte apoptosis [25,26,27,28,29,30]. We, therefore, investigated whether Hippo-YAP signaling is induced by Dox. Cytotoxicity of Dox was demonstrated in HCMs in a dose- and time-dependent manner (Figure 8A). Dox also induced cleavage of caspase-3 and PARP (Figure 8B). As shown in Figure 8C, there was no detectable phosphorylation of MST1/2, LATS1, or YAP in control cells. Dox treatment induced activation of Hippo-YAP signaling resulting in phosphorylated MST1/2, LATS1, and YAP, which resulted in YAP degradation (Figure 8C). Verteporfin (VP), a YAP inhibitor, could induce phosphorylation of YAP and cleavage of caspase-3 and PARP; the combination of VP with Dox (VP+Dox) further increased the levels of phospho-YAP, cleaved caspase-3, and cleaved PARP in HCMs (Figure 8D). These data indicated that the Hippo-YAP pathway is involved in Dox-induced cardiomyocyte apoptosis. To confirm this result in vivo, the mouse heart tissues were analyzed by Western blot and immunofluorescent staining. Similar to our in vitro results, the cardiac tissue from Dox-treated mice showed activated Hippo-YAP signaling, as demonstrated by increased phospho-MST1/2, phospho-LATS1, and phospho-YAP and decreased total YAP and expression of YAP target genes CTGF and CYR61 compared to the control hearts (Figure 8E–G). Ex during therapy suppressed Dox-induced phosphorylation of MST1/2, LATS1, and YAP and preserved total YAP and expression of its target genes CTGF and CYR61. Together, these data demonstrated that activation of the Hippo-YAP pathway is involved in Dox-induced cardiomyocyte apoptosis and may play a role in the development of cardiotoxicity following Dox therapy. Activation of Hippo-YAP signaling was not seen in the Dox+Ex hearts, further confirming that Ex successfully inhibited Dox-induced cardiotoxicity. 

## 4. Discussion

The data presented indicate that Dox therapy damages the cardiac vasculature by killing vascular endothelial cells and pericytes. This in turn results in poor cardiac perfusion, poor tissue oxygenation leading to cardiomyocyte apoptosis, and decreased cardiac function. Exercise during therapy stimulated the migration of BM stem cells which differentiated into endothelial cells and pericytes, thereby repairing the damaged vessels, maintaining blood flow and oxygenation, and preserving cardiac function. Preserving vascular function using an Ex intervention during therapy also inhibited Dox-induced fibroblast proliferation, which is responsible for the development of cardiac fibrosis and impaired contractility. Cardiac fibrosis can compromise function, subsequently leading to the development of heart failure in pediatric and adolescent cancer survivors (Figure 8H).

We also demonstrated that Dox therapy induced activation of Hippo-YAP signaling in the heart, which resulted in cardiomyocyte apoptosis. Similar to the protection of cardiac vessels in the mice that exercised during Dox, the activation of Hippo-Yap signaling and the induction of cardiomyocyte apoptosis were not observed in the Dox+Ex hearts. From these in vivo data, it is unclear whether the cytotoxic effects of Dox therapy were secondary to the decreased blood flow and decreased cardiac oxygenation, which in turn resulted in activation of the Hippo-YAP pathway, cleavage of caspase-3 and PARP, and cardiomyocyte apoptosis, or were mediated by direct activation of Hippo-YAP signaling in addition to the vascular compromise. However, we demonstrated a correlation between cardiomyocyte apoptosis and Hippo-YAP activation in vitro using isolated human cardiomyocytes. Therefore, we interpret this to mean that there is a direct effect of Dox on Hippo-Yap activation that contributed to cardiomyocyte apoptosis. Very interestingly, exercise suppressed Dox-induced activation of Hippo-YAP signaling and apoptosis in the heart and prevented the Dox-induced cardiotoxicity (Figure 8I).

To our knowledge, we are the first to report the findings of damage to the cardiac blood vessels and impaired diastolic and systolic blood flow following Dox therapy. Our study also proposes a unique mechanism of cardiac protection afforded by the use of an exercise intervention during therapy. We demonstrated decreased endothelial cells and pericytes in the cardiac vessels following Dox but not in the cardiac vessels from the mice that exercised during therapy. BM stem cells have the capacity to participate in tissue repair by migrating to the damaged tissue, proliferating, and transdifferentiating into different cell lineages. While a small number of GFP^+^-BM cells was observed in the Dox-treated or control hearts, a significantly higher number of GFP^+^-BM cells was observed in the Dox+Ex-treated mice. Our data suggest that the cardiac vessels in the Dox+Ex-treated mice were repaired by the BM stem cells that were promoted to migrate into the heart and differentiate into endothelial cells and pericytes. 

Vascular repair can involve both angiogenesis, defined as endothelial cell proliferation from pre-existing vasculature, and vasculogenesis, defined as the recruitment of BM-derived precursors into the damaged tissue area with their subsequent differentiation into vascular endothelial cells and pericytes [21,22,24]. Our data indicate that vasculogenesis contributed to the repair of the cardiac vessels injured by Dox therapy. Using a transplant model with GFP^+^ BM cells, we showed that BM cells had significantly migrated into the heart in the Dox+Ex group following therapy and contributed to the endothelial and periendothelial components of the cardiac vasculature. We observed few or very few BM-derived endothelial or periendothelial cells in the control- or Dox-treated hearts. 

Proving that BM cells and vasculogenesis participate in vascular repair following Dox therapy requires a model in which vascular cells derived from the BM can be distinguished from locally derived vascular cells and where only Dox is responsible for the vascular damage. Total body irradiation (TBI) was therefore not used as the preparative regimen prior to transplant. TBI has the potential to damage cardiac cells and has also been shown to inhibit local angiogenesis [43]. We, therefore, elected to use busulfan followed by anti-CD4 and anti-CD8 antibodies as the conditioning regimen prior to transplant with the GFP^+^ BM cells. We recognize that this method does not induce 100% killing of the recipient’s GFP^-^ BM cells; however, our goal was to create a BM where there are GFP-labeled cells. We confirmed engraftment by demonstrating the presence of GFP**^+^** cells in the BM following the transplant using flow cytometry. The strain of both GFP^+^ donor and GFP^-^ recipient mice was C57BL/6. Therefore graft versus host interactions will not occur. Our data show that GFP^+^ endothelial cells and pericytes were significantly present in the hearts from mice treated with Dox+Ex, but few or very few GFP^+^ endothelial cells and pericytes were present in hearts from the control mice or mice treated with Dox alone. We interpret this to mean that Ex stimulated the migration of BM cells into the heart and their subsequent differentiation into vascular cell components.

Pericytes are key mediators of tissue repair and regulate stem cell maintenance. Pericytes also control vessel structure and functionality. Pericytes are critical for vascular stabilization and maintaining an open lumen. Decreased pericyte coverage results in decreased vascular function, decreased tissue perfusion, and increased hypoxia [44,45]. We have previously shown the importance of vasculogenesis and tumor vascular pericyte coverage and the critical role that BM cells and BM-derived pericytes play in the expansion of tumor vasculature [19,20,21,23,24]. Inhibiting vasculogenesis and BM stem cell migration decreased vascular perfusion, increased tumor tissue hypoxia and apoptosis, and compromised the tumor’s ability to recover and regrow following chemotherapy or radiation therapy as the oxygen and nutrients required for recovery were not delivered [46]. By contrast, stimulating this process aided in recovery following antiangiogenic therapy [22]. Cardiomyocyte apoptosis and cell death were documented in the Dox-treated mice but not in the Dox+Ex-treated mice. We interpreted this finding to mean that the compromised cardiac blood flow (resulting from decreased pericyte coverage) contributed to Dox-induced apoptosis. Therefore, our studies are novel as we have identified a new mechanism for Dox-induced cardiotoxicity and a mechanism by which exercise inhibits Dox-induced vascular damage and cardiomyocyte apoptosis, both of which are responsible for cardiotoxicity. Exercise not only preserved cardiac blood flow but also protected against the proliferation of fibroblasts, which presumably lead to cardiac fibrosis and compromised contractility.

Intensified Dox therapy has significantly improved the survival of sarcoma patients. However, the price of this dose intensification is the development of early and late cardiotoxicities. Up to 57% of survivors of childhood sarcoma experience late-onset cardiotoxicity and heart failure, and pediatric cancer survivors are 8.2 times more likely to experience premature death from heart disease than their siblings [47,48]. Two mechanisms have previously been identified that mediate Dox-induced cardiac damage. One is through the generation of iron-mediated free radicals that induce myocardial cell apoptosis [8,9]. The other is thought to involve the binding of topoisomerase-IIβ, which is abundant in cardiomyocytes, leading to the inhibition of DNA replication. This in turn increases cardiomyocyte apoptosis [9,10,11]. Our data add to this body of knowledge. We are the first to show that in addition to the above mechanisms, Dox-induced cardiac damage may also be mediated by an effect on the cardiac vessels that leads to decreased blood flow in the heart and that the apoptotic process is controlled by the Hippo-YAP signaling pathway. In addition, we demonstrated that exercise during Dox therapy prevented both the vascular damage and induction of the Hippo-YAP signaling and preserved blood flow and cardiac function. Our transplant studies suggest that this preservation of cardiac blood flow was mediated by Ex-induced migration of c-kit^+^ BM stem cells into the heart; these c-kit^+^ BM stem cells subsequently differentiated into vascular cells, thereby repairing the damaged vessels and maintaining normal systolic and diastolic blood flow. Our studies also confirm the findings of Fazel et al., which showed that the recruitment of c-kit^+^ BM cells is important in the cardiac repair process by establishing a proangiogenic niche in infarcted myocardium [49,50]. Similar to our studies, the recruited c-kit^+^ cells did not differentiate into cardiomyocytes but rather initiated and enhanced cardiac repair and recovery following myocardial infarction by initiating angiogenesis [49,50]. Therefore, the Ex-induced cardiac repair mechanism may not be unique but may also be induced by other interventions that promote cardiac healing following infarction or vascular injury.

In summary, we believe that in addition to inducing the generation of reactive oxygen species and inhibiting DNA replication, Dox therapy damages the cardiac vessels by killing vascular endothelial cells and pericytes. This leads to decreased cardiac blood flow and decreased tissue oxygenation, which in turn contribute to cardiomyocyte death and proliferation of fibroblasts. Furthermore, our data show that activation of the Hippo-YAP signaling pathway correlates with Dox-induced cardiotoxicity. Therefore, in addition to combining exercise with Dox treatment, identifying agents that inhibit Hippo-YAP signaling may offer another approach to protect the heart, decrease cardiotoxicity, and preserve cardiac function in survivors. Such interventions during frontline therapy may provide cardiac protection, thereby reducing the incidence of Dox-associated cardiovascular disease in childhood and adolescent sarcoma survivors.

## 5. Conclusions

In summary, this study demonstrated that exercise inhibited Dox-induced cardiomyocyte apoptosis and vascular damage. We uncovered the mechanisms involved in exercise-mediated amelioration of Dox-induced cardiotoxicity. Exercise promotes migration of BM stem cells into the heart and their differentiation into endothelial cells and pericytes, and this process may be involved in the repair of cardiac blood vessels damaged by Dox. Dox-induced cardiomyocyte apoptosis involves activation of the Hippo-YAP signaling pathway, and exercise inhibits Dox-induced activation of the Hippo-YAP pathway and apoptosis. This study demonstrated that exercise decreased acute Dox-induced cardiac damage and, therefore, can be used as an intervention to reduce cardiac disease and heart failure in childhood and adolescent cancer survivors, particularly those who received high doses of Dox. Targeting Hippo-YAP signaling might be also an effective therapeutic strategy to protect the heart from Dox-induced heart damage.

## Figures and Tables

**Figure 1 cancers-13-02740-f001:**
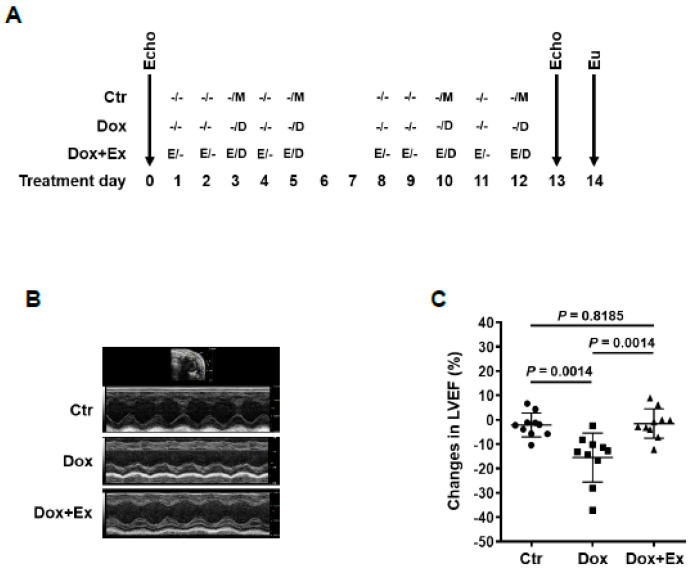
Exercise prevents Dox-induced decreases in cardiac function and blood flow. (**A**) Schematic representation of experiment design and mouse treatment. (**B**,**E**,**G**) Representative Echo images of M-mode of parasternal short axis (PSAX) view of hearts (**B**), PW Doppler waveform of trans-mitral blood flow in hearts (**E**), and PW Doppler waveform of transverse aorta blood flow in hearts (**G**) from Ctr, Dox, and Dox+Ex group mice. (**C**,**D**,**F**,**H**) Echo data show changes in LVEF (**C**), LVFS (**D**), mean velocity at LVMV (**F**), and mean velocity at LVOT (**H**) after treatment in Ctr, Dox, and Dox+Ex group mice. N = 10 mice/group. *P* values are indicated by the GraphPad *t* test. Ctr vs Dox+Ex, not significant statistically. (**I**–**K**) Echo data show changes in LVEF (**I**), LVFS (**J**), and mean velocity at LVMV (**K**) after treatment in Ctr and Ex group mice. N = 8 mice/group. *P* values are indicated by the GraphPad *t* test. Ctr vs Ex, not significant statistically. LVEF, left ventricular ejection fraction; LVFS, left ventricular fractional shortening; LVMV, left ventricular mitral valve; LVOT, left ventricular outflow tract in the ascending aorta; PW Doppler, pulse wave Doppler; Ctr, control; M, mock; Ex/E, exercise; Dox/D, doxorubicin; Echo, echocardiography; Eu, euthanize.

**Figure 2 cancers-13-02740-f002:**
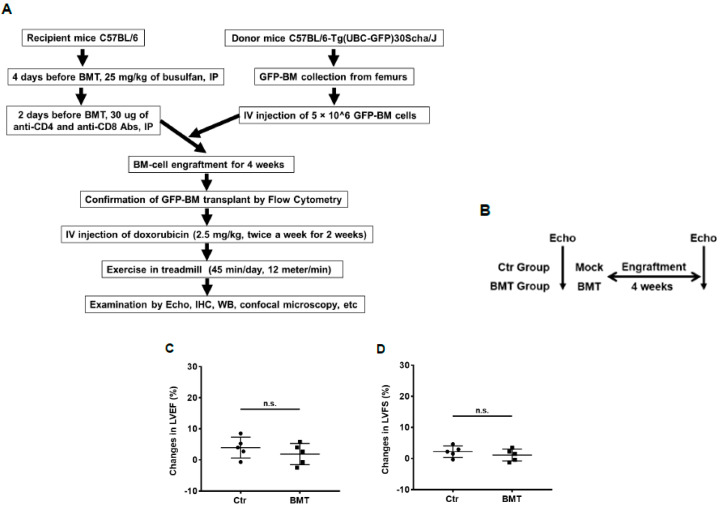
Bone marrow transplantation does not affect cardiac function and blood flow. (**A**) Schematic representation of bone marrow transplantation (BMT) procedure and subsequent mouse treatment. (**B**) Schematic representation of BMT experiment design. (**C**–**H**) Echo data show the difference between Ctr and BMT group mice in LVEF, LVFS, MV E/A Ratio, MV ET, LVAW;d, and LVPW;d. N = 5 mice/group. n.s. indicates not statistically significant by the GraphPad *t* test. LVEF, left ventricular ejection fraction; LVFS, left ventricular fractional shortening; MV E/A Ratio, mitral valve peak E and A velocity ratio; MV ET, mitral valve ejection time; LVAW;d, left ventricular anterior wall thickness in diastole; LVPW;d, left ventricular posterior wall thickness in diastole; Ctr, control; BMT, bone marrow transplantation.

**Figure 3 cancers-13-02740-f003:**
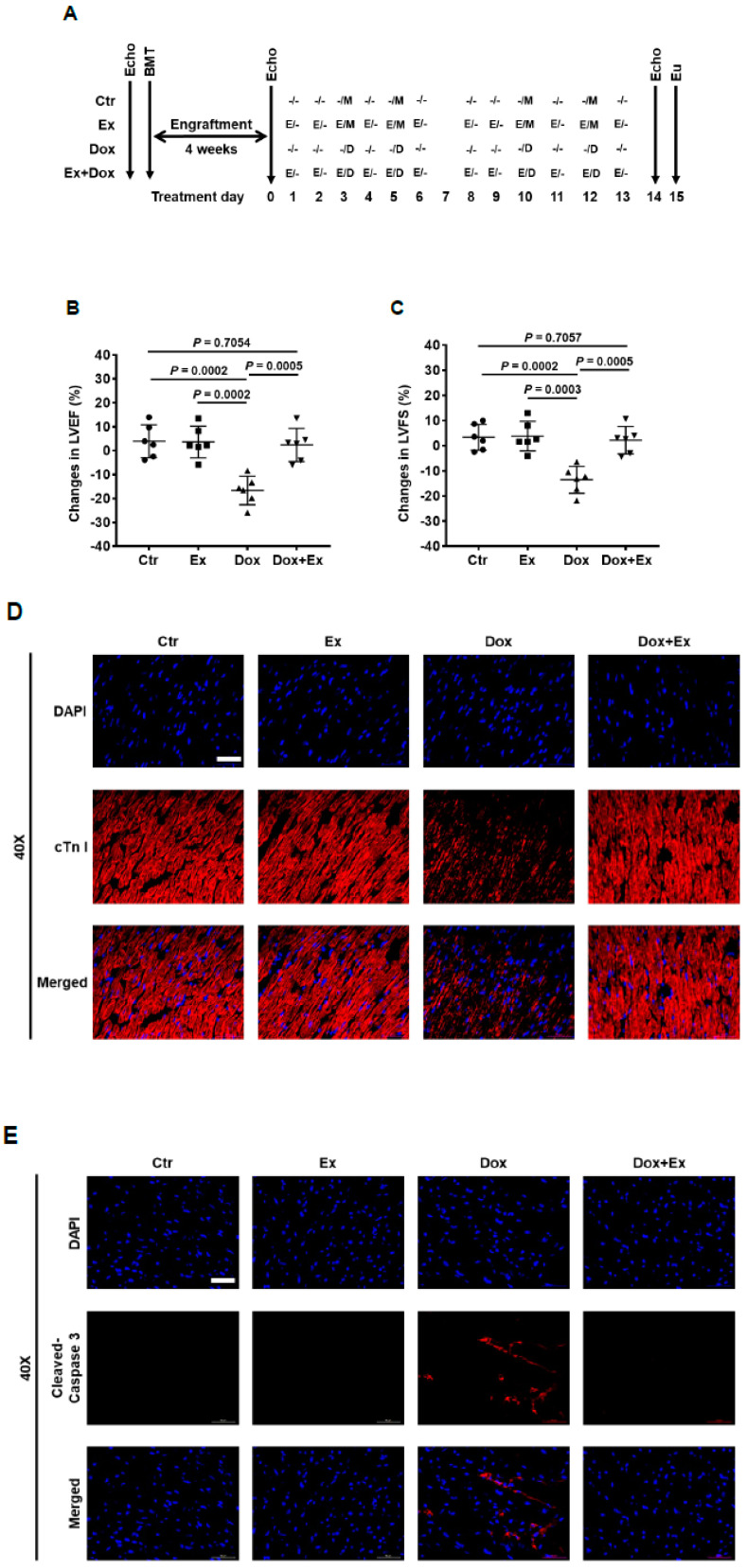
Exercise inhibits Dox-induced cardiomyocyte apoptosis. (**A**) Schematic representation of experiment design of BMT and mouse treatment. (**B**,**C**) Echo data show changes in LVEF and LVFS before and after treatment in Ctr, Ex, Dox, and Dox+Ex group mice. N = 6 mice/group. *P* values are indicated by the GraphPad *t* test. Ctr vs Dox+Ex, not statistically significant. (**D**) Representative immunofluorescence images of heart sections stained for DAPI (blue) and cTn I (red) from Ctr, Ex, Dox, and Dox+Ex group mice. Magnification, 40×; Scale bar, 50 µm. (**E**) Representative immunofluorescence images of heart sections stained for DAPI (blue) and cleaved caspase-3 (red) from Ctr, Ex, Dox, and Dox+Ex group mice. Magnification, 40×; Scale bar, 50 µm. (**F**) Western blot analysis for cleaved caspase-3, cleaved PARP, and histone H3 expression in hearts from Ctr, Ex, Dox, and Dox+Ex group mice. LVEF, left ventricular ejection fraction; LVFS, left ventricular fractional shortening; Ctr, control; M, mock; Ex/E, exercise; Dox/D, doxorubicin; Echo, echocardiography; Eu, euthanize; BMT, bone marrow transplant.

**Figure 4 cancers-13-02740-f004:**
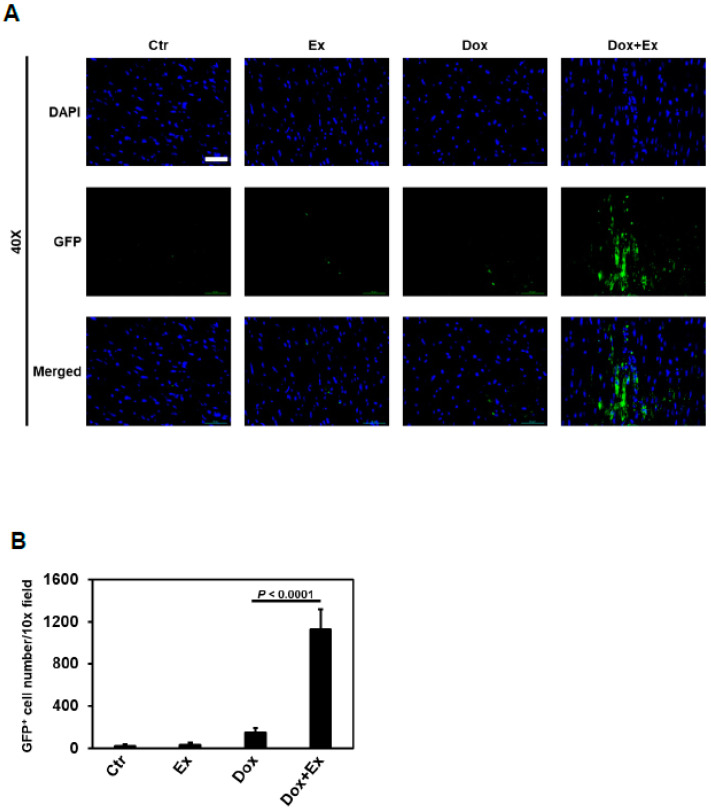
Exercise promotes migration of bone marrow (stem) cells into the heart after Dox treatment. (**A**) Representative immunofluorescence images of heart sections stained for DAPI (blue) and GFP (green) from Ctr, Ex, Dox, and Dox+Ex group mice. Magnification, 40×; Scale bar, 50 µm. (**B**) Quantification of GFP-positive cells in heart tissues per 10× field from Ctr, Ex, Dox, and Dox+Ex group mice. N = 5 mice/group, 3 slides/mouse. Values represent mean ± SEM. *P* values are indicated by the Student’s *t* test. (**C**) Representative immunofluorescence images of heart sections stained for DAPI (blue), GFP (green), and c-Kit (red) from Ctr, Ex, Dox, and Dox+Ex group mice. Magnification, 400×; Scale bar, 5 µm. (**D**,**E**) Quantification of the number and percentage of c-Kit and GFP double-positive cells in heart tissues per 10× field and per total c-Kit-positive cells from Ctr, Ex, Dox, and Dox+Ex group mice. N = 5 mice/group, 3 slides/mouse. Values represent mean ± SEM. *p* values are indicated by the Student’s *t* test.

**Figure 5 cancers-13-02740-f005:**
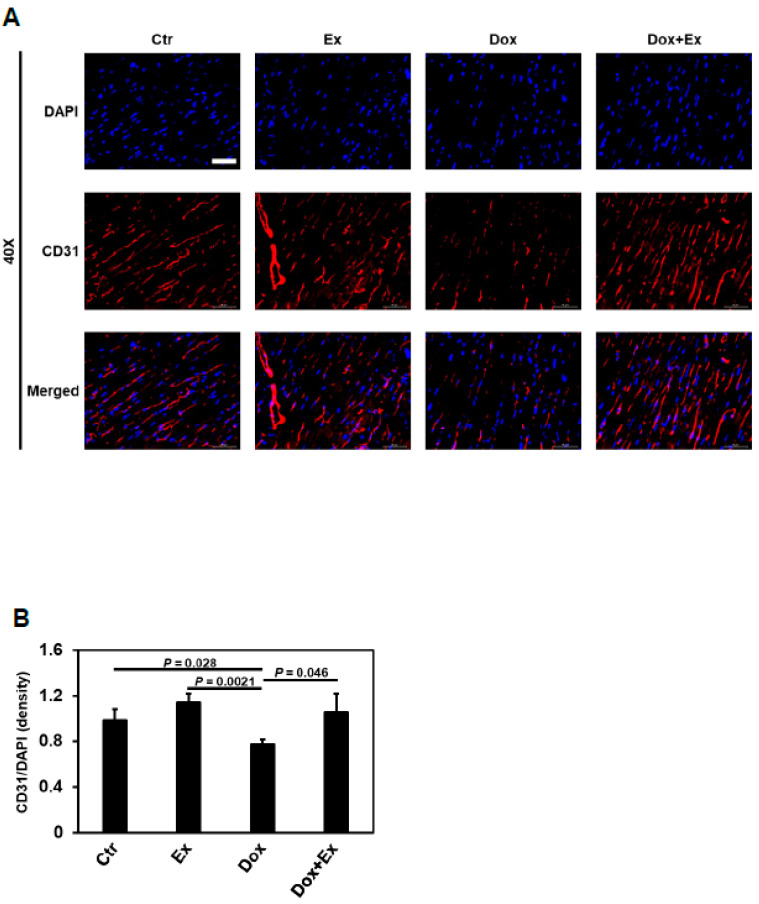
Exercise promotes differentiation of BM stem cells into endothelial cells after Dox treatment. (**A**) Representative immunofluorescence images of heart sections stained for DAPI (blue) and CD31 (red) from Ctr, Ex, Dox, and Dox+Ex group mice. Magnification, 40×; Scale bar, 50 µm. (**B**) Quantification of CD31-positive cells normalized to DAPI (density) in heart tissues from Ctr, Ex, Dox, and Dox+Ex group mice. N = 5 mice/group, 3 slides/mouse. Values represent mean ± SEM. *P* values are indicated by the Student’s *t* test. Ctr vs Dox+Ex, not statistically significant. (**C**) Representative immunofluorescence images of heart sections stained for DAPI (blue), GFP (green), and CD31 (red) from Ctr, Ex, Dox, and Dox+Ex group mice. Arrows indicate colocalization (yellow) of GFP (green) and CD31 (red). Magnification, 40×; Scale bar, 50 µm. (**D**) Quantification of CD31 and GFP double-positive cells in heart tissues per 10× field from Ctr, Ex, Dox, and Dox+Ex group mice. N = 5 mice/group, 3 slides/mouse. Values represent mean ± SEM. *P* values are indicated by the Student’s *t* test. (**E**) Quantification of percentage of CD31 and GFP double-positive cells to total CD31-positive cells in heart tissues from Ctr, Ex, Dox, and Dox+Ex group mice. N = 5 mice/group, 3 slides/mouse. Values represent mean ± SEM. *p* values are indicated by the Student’s *t* test.

**Figure 6 cancers-13-02740-f006:**
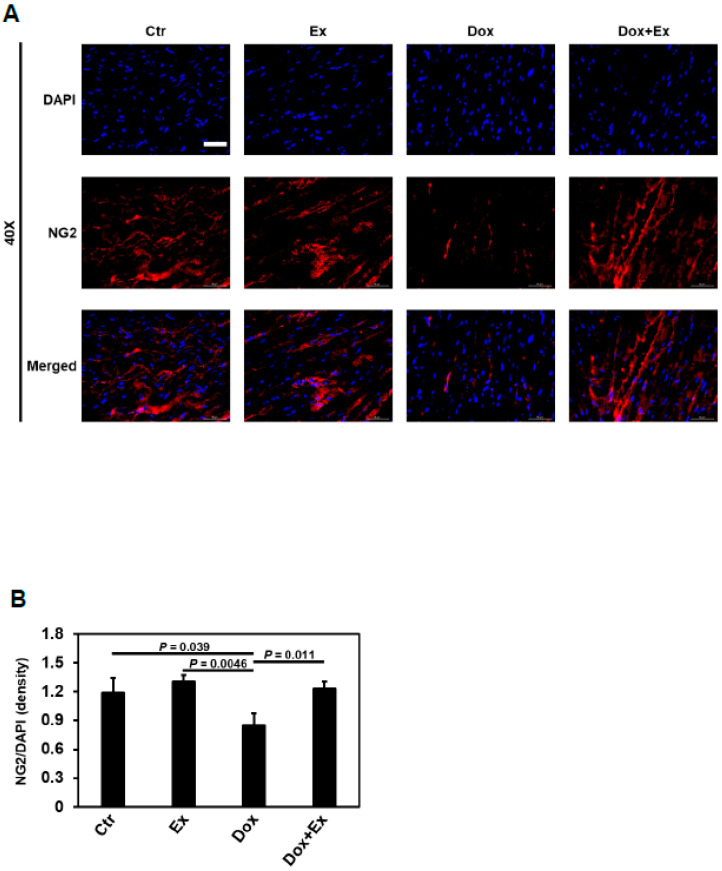
Exercise promotes differentiation of BM stem cells into pericytes after Dox treatment. (**A**) Representative immunofluorescence images of heart sections stained for DAPI (blue) and NG2 (red) from Ctr, Ex, Dox, and Dox+Ex group mice. Magnification, 40×; Scale bar, 50 µm. (**B**) Quantification of NG2-positive cells normalized to DAPI (density) in heart tissues from Ctr, Ex, Dox, and Dox+Ex group mice. N = 5 mice/group, 3 slides/mouse. Values represent mean ± SEM. *P* values are indicated by the Student’s *t* test. Ctr vs Dox+Ex, not statistically significant. (**C**) Representative immunofluorescence images of heart sections stained for DAPI (blue), GFP (green), and NG2 (red) from Ctr, Ex, Dox, and Dox+Ex group mice. Arrows indicate colocalization (yellow) of GFP (green), and NG2 (red). Magnification, 40×; Scale bar, 50 µm. (**D**) Quantification of NG2 and GFP double-positive cells in heart tissues per 10× field from Ctr, Ex, Dox, and Dox+Ex group mice. N = 5 mice/group, 3 slides/mouse. Values represent mean ± SEM. *p* values are indicated by the Student’s *t* test. (**E**) Quantification of percentage of NG2 and GFP double-positive cells to total NG2-positive cells in heart tissues from Ctr, Ex, Dox, and Dox+Ex group mice. N = 5 mice/group, 3 slides/mouse. Values represent mean ± SEM. *p* values are indicated by the Student’s *t* test.

**Figure 7 cancers-13-02740-f007:**
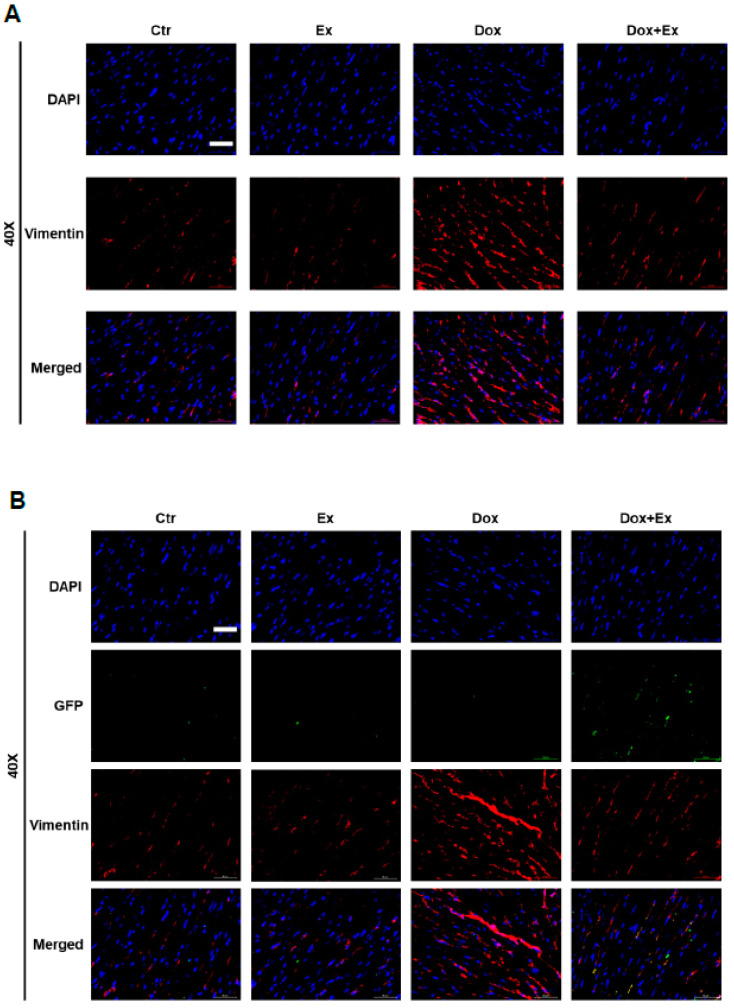
Exercise prevents Dox-induced increase in fibroblasts. (**A**) Representative immunofluorescence images of heart sections stained for DAPI (blue) and Vimentin (red) from Ctr, Ex, Dox, and Dox+Ex group mice. Magnification, 40×; Scale bar, 50 µm. (**B**) Representative immunofluorescence images of heart sections stained for DAPI (blue), GFP (green), and Vimentin (red) from Ctr, Ex, Dox, and Dox+Ex group mice. Magnification, 40×; Scale bar, 50 µm.

**Figure 8 cancers-13-02740-f008:**
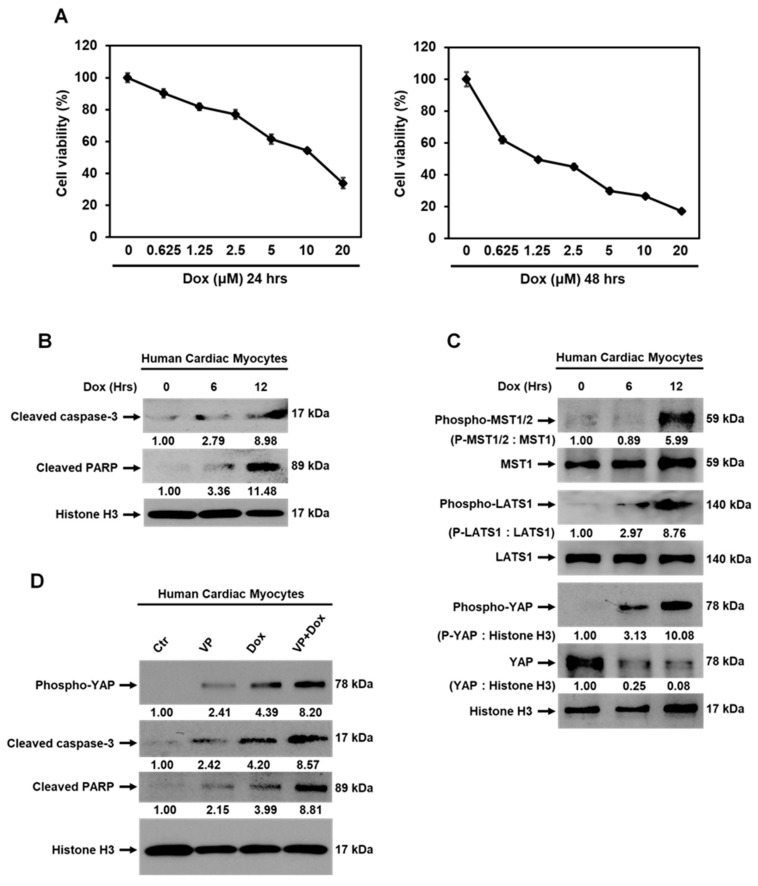
Activation of Hippo-YAP signaling is induced by Dox therapy. (**A**) MTT assay analysis for viability of human cardiac myocytes (HCMs) treated with various concentrations of Dox as indicated for 24 or 48 h. Values represent mean ± SEM of three independent experiments. (**B**) Western blot analysis for cleaved caspase-3, cleaved PARP, and histone H3 expression in HCMs treated with Dox at 1.25 µM for 0, 6, or 12 h. (**C**) Western blot analysis of phospho-MST1/2, MST1, phospho-LATS1, LATS1, phospho-YAP, YAP, and histone H3 expression in HCMs treated with Dox at 1.25 µM for 0, 6, or 12 h. (**D**) Western blot analysis of phospho-YAP, cleaved caspase-3, cleaved PARP, and histone H3 expression in HCMs treated with mock (control), verteporfin (VP) at 2.5 µM, Dox at 1.25 µM, or combination of VP with Dox for 8 h. (**E**,**F**) Western blot analysis of phospho-MST1/2, MST1, phospho-LATS1, LATS1, phospho-YAP, YAP, histone H3 (**E**), CTGF, CYR61, and Histone H3 (**F**) expression in hearts from Ctr, Ex, Dox, and Dox+Ex group mice. (**G**) Representative immunofluorescence images of heart sections stained for DAPI (blue) and p-YAP (red) from Ctr, Ex, Dox, and Dox+Ex group mice. Magnification, 40×; Scale bar, 50 µm. (**H**,**I**) Summary of the cellular and molecular mechanisms of Ex-mediated protection of Dox-induced cardiotoxicity as described in Section 4. Ex, exercise; Dox, doxorubicin; VP, verteporfin; CTGF, connective tissue growth factor; CYR61, cysteine-rich angiogenic inducer 61.

## Data Availability

All data generated and/or analyzed in this study are included in this article and its Appendix A files and are available from the corresponding author upon reasonable request.

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
