# Peer review of "Exercise Inhibits Doxorubicin-Induced Damage to Cardiac Vessels and Activation of Hippo/YAP-Mediated Apoptosis"

_cancers, 2021, doi:10.3390/cancers13112740_

Round 1

Reviewer 1 Report

The manuscript has been implemented as suggested.

Author Response

We thank the reviewer for accepting our revised manuscript. 

Thank you so much!

Reviewer 2 Report

Despite most of my comments have been correctly addressed, I still think point 2 should be fixed. Moreover, quality of some immunoblot are not good and they are difficult to quantify. For example, Histone H3, Phospho-YAP. A better immunoblot image should be included.

  1. What is the effect of Ex in cardiovascular function and blood flow? Why this group was not included in fig 1? Please see the data in Fig. 3B, 3C, S3A-S3J, S15A, and S15B. When compared to the control, there was no significant effect of Ex alone on cardiovascular function and blood flow. These experiments were done before Fig.1. Based on this information, we didn’t include the Ex group in Fig. 1. I still think Ex group should be included in figure 1.

Author Response

We thank the reviewer for accepting most of our revisions. For the below two questions, we have addressed and made revisions to the manuscript accordingly. We appreciate the reviewer’s kind consideration.

Comments and Suggestions for Authors

Despite most of my comments have been correctly addressed, I still think point 2 should be fixed. Moreover, quality of some immunoblot are not good and they are difficult to quantify. For example, Histone H3, Phospho-YAP. A better immunoblot image should be included.

As requested, we used the original samples and did the western blots, and better immunoblot images for Histone H3 and Phospho-YAP have been included. Please kindly see these data in Fig. 3F on page 13, Fig. 8B and 8C on page 25, and Fig. 8E on page 26.

What is the effect of Ex in cardiovascular function and blood flow? Why this group was not included in fig 1?

Please see the data in Fig. 3B, 3C, S3A-S3J, S15A, and S15B. When compared to the control, there was no significant effect of Ex alone on cardiovascular function and blood flow. These experiments were done before Fig.1. Based on this information, we didn’t include the Ex group in Fig. 1.

I still think Ex group should be included in figure 1.

We thank the reviewer for this question. The Echo data showing the effect of Ex on cardiovascular function and blood flow have been included in Figure 1. Please kindly see these data in Fig. 1I-1K on page 7. These data indicated that Ex alone had no significant effect on cardiovascular function and blood flow. These results are consistent with our data shown in Fig. 3B, 3C, S3A-S3J, S15A, and S15B; and are also consistent with our lab member Dr. Fei Wang’s data showing that there are no significant differences in cardiovascular function and blood flow between control and Ex alone group mice in her manuscript which is under revision.

Round 2

Reviewer 2 Report

My comments have been correctly addressed

This manuscript is a resubmission of an earlier submission. The following is a list of the peer review reports and author responses from that submission.

Round 1

Reviewer 1 Report

The manuscript from Tao et al., is well descripted and organized. The research design is appropriate.

Since cardiac damage doxorubicin-induced is a relevant issue in cancer patients, these results can contribute to clarify mechanisms to prevent damage.

I suggest some actions to improve the manuscript

  • To confirm the Yap pathway involvement, I suggest to test a common Yap inhibitor, such as Verteporfin, alone and in combination with doxorubicin, in cardiac myocytes on p-YAP, parp and casp 3
  • I suggest to move in supplementary Fig.4 (F-G-H-I-J-L) Fig.5 (F-G-H-I-J-) Fig 6. (F-G-H-I-J-)

Author Response

We thank the reviewer for providing the helpful suggestions and comments. We have revised the manuscript in accordance with these comments.

The manuscript from Tao et al., is well descripted and organized. The research design is appropriate.

Since cardiac damage doxorubicin-induced is a relevant issue in cancer patients, these results can contribute to clarify mechanisms to prevent damage.

I suggest some actions to improve the manuscript

1. To confirm the Yap pathway involvement, I suggest to test a common Yap inhibitor, such as Verteporfin, alone and in combination with doxorubicin, in cardiac myocytes on p-YAP, parp and casp 3.

As requested, we have performed additional experiments to address this point. Cardiac myocytes were treated with Mock (control), Verteporfin (VP) alone, Doxorubicin (Dox) alone, or VP+Dox. The levels of p-YAP, cleaved PARP and cleaved caspase-3 were then examined by western blot. We found that compared to the control, VP alone and Dox alone induced phosphorylation of YAP and cleavage of PARP and caspase-3. Combination of VP with Dox (VP+Dox) further increased the levels of p-YAP, cleaved PARP and cleaved caspase-3. These results confirm that the YAP pathway is involved in Dox-induced cardiomyocyte apoptosis. Please see these new data in Figure 8D. Page 23 line 506-509, Page 24 fig. D, Page 26 line 534-535.

2. I suggest to move in supplementary Fig.4 (F-G-H-I-J) Fig.5 (F-G-H-I-J) Fig 6. (F-G-H-I-J).

As requested, we have moved these figures to our supplementary materials. Please see these figures in Figure S7(A-D), S8, S11(A-E), and S14(A-E). Page 15 line 373-379, Page 18 line 421-426, Page 21 line 462-467, Supple page 12 line 18-22, Supple page 13 line 1-9 and line21-23, Supple page 14 line 1-8 and line 20-23, Supple page 15 line 1-7.

Reviewer 2 Report

Overall, this is an interesting manuscript, as the authors reported that activation of the Hippo-YAP signaling pathway correlates with Dox-induced cardiotoxicity, and exercise inhibits Dox-induced activation of the Hippo-YAP pathway and apoptosis.  These results can be important to increase our knowledge about molecular mechanisms of Dox-induced cardiovascular damage. However, some issues should be addressed to be this manuscript more suitable for publication in Cancers.

  1. How the authors selected the Exercise (Ex) and Doxorubicin (Dox) Treatment? Why 2 weeks (total 10 or 12 days of exercise)?. Can regular exercise practice before Dox treatment also prevent Dox-induced cardiovascular damage or Dox-induced cardiovascular damage decrease more with concomitant exercise practice? Are the young athletes more resistant to  Dox-induced cardiovascular damage?
  2. What is the effect of Ex in cardiovascular function and blood flow? Why this group was not included in fig 1?
  3. Statistic test and samples number are not included in figure legends. Authors should include these information in all figures.
  4. Figures 4, 5 and 6 have too much information. The authors should simplified and include only the information relevant in the main figures and add new suppl figures. 
  5. How the immunoblot were performed? Why some immunoblot look very bad? There are many incomplete bands, thin, dotty lanes and not separation between lanes. In my opinion all these troubleshooting can difficult the immunoblot quantification. A better immunoblot imagen should be included.
  6. Exercise suppressed Dox-induced activation of Hippo-YAP signaling and apoptosis in the heart. Do the authors any information of the effect on expression levels of selected YAP target genes?
  7. I suggest that authors add all immunoblot to supplementary material, not only the representative one shows in the figures.

Author Response

We thank the reviewer for providing the important suggestions and comments. We have made revisions to the manuscript to address all of the questions. These are detailed below.

Overall, this is an interesting manuscript, as the authors reported that activation of the Hippo-YAP signaling pathway correlates with Dox-induced cardiotoxicity, and exercise inhibits Dox-induced activation of the Hippo-YAP pathway and apoptosis.  These results can be important to increase our knowledge about molecular mechanisms of Dox-induced cardiovascular damage. However, some issues should be addressed to be this manuscript more suitable for publication in Cancers.

  1. How the authors selected the Exercise (Ex) and Doxorubicin (Dox) Treatment? Why 2 weeks (total 10 or 12 days of exercise)?

We selected the Dox dosing regimen used in these studies based on the dose conversion investigations between mice and humans published by Nair and Jacob (Nair AB, Jacob S. A simple practice guide for dose conversion between animals and human. J Basic Clin Pharm. 2016;7:27–31.) This study described a method to convert animal dosing (in mg/kg) to human equivalents (in mg/kg) by dividing the mouse dose (in mg/kg) by 12.3. The 2.5 mg/kg dose given twice a week for 2 weeks is roughly equivalent to the dose that is used in pediatric patients. Therefore, for our model, 4 doses over 2 weeks at 2.5 mg/kg/dose equals a human equivalent dose of 0.813 mg/kg; 10 mg/kg divided by 12.3. From CDC growth charts, we then calculated the surface area for an average 5-year old-male based on 50th percentile for height and weight as follows: Weight (18 kg) × height (109 cm)/3600 =0.55 m2. Using the dose of Dox delivered during delayed intensification in the current Children’s Oncology Group high-risk ALL study (AALL1131) of 25 mg/m2, the absolute dose of Dox given is 13.75 mg/dose; 25 mg/m2×0.55m2= 13.8 mg. This compares with a mouse dose converted to human equivalent dosing of 14.6 mg; 0.813 mg/kg×18 kg=14.6 mg.

We used a moderate intensity exercise regimen (12 m/min, equivalent to ~65% VO2 max for mice, ~brisk walk for humans) and not a high intensity or exhaustive regimen as it is unlikely that pediatric cancer patients can do a high intensity regimen, sustain such a regimen over an extended period of time and finally because of the location of osteosarcoma (most commonly in the femur and tibia), the orthopedic surgeons discourage any exercise with high impact to the legs, as this can result in bone fracture because the tumor weakens the bone. Moderate walking is acceptable. We did, however, allow animals to acclimate to the treadmill each session by beginning the treadmill at a low speed and slowly increasing to the exercise speed that was used for each 45 minutes session.

Animals were exercised for 2 weeks (total 10 or 12 days of exercise) because these animals were treated with 2.5 mg/kg of Dox, twice a week for 2 weeks, exercise matched the Dox treatment time.

Animals were not pretrained. This was an intentional choice because pretraining animals may cause changes to cardiac fitness that would skew our results. The average pediatric patient in the United States does not meet national guidelines for weekly physical activity, meaning that the patient population that we are trying to model is not pretrained before a cancer diagnosis. Therefore, mice that are not pretrained make the most appropriate model to study the impact of exercise during Dox therapy.

Can regular exercise practice before Dox treatment also prevent Dox-induced cardiovascular damage or Dox-induced cardiovascular damage decrease more with concomitant exercise practice?

We have not examined whether exercise conditioning before treatment impact Dox-induced cardiotoxicity. This is indeed a reasonable suggestion as this is being investigated with breast cancer patients. These patients usually have time before the initiation of chemotherapy as most undergo surgery and lymph-node analysis before chemotherapy is imitated so there is time for exercise preconditioning. However, for children with osteosarcoma, pre-operative chemotherapy begins immediately once the diagnosis is made and before surgery is performed so there is no time to initiate a precondition exercise regimen. Since our focus in children and adolescents with osteosarcoma and other sarcomas, we elected to investigate the use of an exercise intervention during therapy.

Are the young athletes more resistant to Dox-induced cardiovascular damage? 

There is no data to date that shows that physical fitness at diagnosis correlates with less cardiac late effects in childhood cancer survivors.

  1. What is the effect of Ex in cardiovascular function and blood flow? Why this group was not included in fig 1?

Please see the data in Fig. 3B, 3C, S3A-S3J, S15A, and S15B. When compared to the control, there was no significant effect of Ex alone on cardiovascular function and blood flow. These experiments were done before Fig. 1. Based on this information, we didn’t include the Ex group in Fig. 1.

  1. Statistic test and samples number are not included in figure legends. Authors should include these information in all figures.

The statistical methods are described in the “Methods” section and the sample numbers are described in the figure legends. We have now also included the statistic tests behind the sample numbers in all figure legends. Page 7 line 261, Page 9 line 293, Page 12 line 314, Page 15 line 369 and 372, Page 17 line 415, Page 18 line 419 and 421, Page 20 line 456, Page 21 line 460 and 462, Supple page 11 line 5, 11 and 19, Supple page 12 line 10 and 16, Supple page 13 line 19, Supple page 14 line 18, Supple page 15 line 12.

  1. Figures 4, 5 and 6 have too much information. The authors should simplified and include only the information relevant in the main figures and add new suppl figures.

In accordance with the reviewer’s suggestion, we have moved Figure 4(F-J), Figure 5(F-J), and Figure 6(F-J) to our supplementary materials. Please see these figures in Figure S7(A-D), S8, S11(A-E), and S14(A-E). Page 15 line 373-379, Page 18 line 421-426, Page 21 line 462-467, Supple page 12 line 18-22, Supple page 13 line 1-9 and line21-23, Supple page 14 line 1-8 and line 20-23, Supple page 15 line 1-7.

  1. How the immunoblot were performed? Why some immunoblot look very bad? There are many incomplete bands, thin, dotty lanes and not separation between lanes. In my opinion all these troubleshooting can difficult the immunoblot quantification. A better immunoblot imagen should be included.

We performed the western blot according to the detailed protocol described in our supplementary materials. For this immunoblot, the samples were from mouse heart tissues which comprise multiple different types of cells. A fresh stronger lysis buffer is better to lyse the mouse heart tissue. We have now attached the whole immunoblots in our supplementary material-western blot file.

  1. Exercise suppressed Dox-induced activation of Hippo-YAP signaling and apoptosis in the heart. Do the authors any information of the effect on expression levels of selected YAP target genes?

This is an interesting point. To investigate the effect of Exercise (Ex) and Doxorubicin (Dox) on expression levels of selected YAP target genes, we used the heart tissues from Control (Ctr), Ex, Dox, and Dox+Ex mice, performed the western blot experiments, and then examined the expression levels of YAP target genes CTGF and CYR61, two anti-apoptosis genes. These data showed that Dox treatment decreased the expression levels of YAP target genes CTGF and CYR61. This was not seen in the heart tissue from the Dox+Ex-treated mice. These data are consistent with our results in Fig. 8E showing that Dox  but not Dox+Ex treatment decreased total YAP expression level. Please kindly see these data in Fig. 8F. Page 23 line 514 and 516, Page 25 fig. F, Page 26 line 536-537.

  1. I suggest that authors add all immunoblot to supplementary material, not only the representative one shows in the figures.

We thank the reviewer for this suggestion. We have added all the immunoblots to the supplementary materials. Please now find them in the supplementary material-western blot file.